# Study of the Relationship between Sensory Processing Sensitivity and Burnout Syndrome among Speech and Language Therapists

**DOI:** 10.3390/ijerph20237132

**Published:** 2023-12-01

**Authors:** Jimmy Bordarie, Caroline Mourtialon

**Affiliations:** 1Qualité de vie et Santé Psychologique, University of Tours, 37041 Tours, France; 2APHP Hôpital Sainte-Périne, 75016 Paris, France; caroline.mourtialon@aphp.fr

**Keywords:** sensory processing sensitivity, burnout, quality of life, vulnerability, occupational health, speech language therapists

## Abstract

Background: Burnout syndrome can arise due to either situational factors such as working conditions, or dispositional factors such as certain temperaments, like a high sensory processing sensitivity. We aim to address the relative absence of studies on speech-language therapists and seek to determine the role of high sensitivity for these healthcare workers in relation to burnout syndrome. Method: The sample consisted of 602 female speech-language pathologists who anonymously completed a questionnaire measuring burnout (ProQOL 5th edition) and sensory processing sensitivity (HSPS-FR). Results: The results revealed that 77.41% of the participants reported moderate or high burnout scores. Furthermore, the analyses revealed that highly sensitive participants are more vulnerable to burnout than others. Conclusion: This study highlights the negative impact of high sensory processing sensitivity on burnout. However, according to the kind of sensitivity, we discuss the way in which highly sensitive caregivers can master this sensitivity in order to use it as a strength in their professions and to spare themselves from suffering deleterious effects, such as compassion fatigue and/or burnout.

## 1. Introduction

Burnout (BO) syndrome is defined as a psychological reaction to prolonged occupational stress that is influenced by both the individual and the organisational context [1]. It has been widely studied since Maslach and Jackson [2,3] first theorised it. In France, according to prevalence studies, BO could affect up to 34% [4] of the French working population, with 13% suffering from high burnout. The most vulnerable groups appear to be women (44% suffer from psychological distress) and those under 29 years of age (59%). There is an extensive literature focus in particular on BO among healthcare professionals due to its multiple consequences such as job dissatisfaction, intention to change jobs or professions [5], increased anxiety, depression, or even addiction [6], or its negative effects on patient care [7]. Healthcare workers are a particularly vulnerable group, with a higher prevalence of the so-called “caregiver burnout syndrome” (CSBS) [8] than the general population. More specifically, self-employed nurses are the most affected by this syndrome (56.5%), along with speech therapists (48.6%), chiropodists (48.5%), orthoptists (39.8%), and physiotherapists (32%) [9].

This syndrome may be due to various organisational factors, such as poor interpersonal relations and a heavy workload [10] or its uneven distribution and unstable team dynamics [11,12]. Some demographic variables also seem to play a role, revealing higher levels of BO for individuals who are young, female, and with a lower level of education than others [13,14]. Finally, CSBS is characterised by certain specificities, notably due to the emotional demands of these professions and the suffering that carers can be subjected to [15], leading to a dehumanisation of the relationship with patients [16]. Some studies are interested in other factors that may explain BO, notably through the study of specific qualities or dispositional variables. Indeed, among the qualities expected of a career, certain characteristics such as empathy [17], involvement, awareness, and sensitivity [18] are thought to heighten the appearance of BO [16,19]. Some researchers are also interested in high sensory sensitivity as a factor of vulnerability [20,21].

High sensitivity to sensory processing can be defined as a stable temperamental trait [22] affecting about 30% of the population [23]. This trait allows individuals to adapt more easily to changing environmental conditions [24]. A highly sensitive individual is characterised by a higher level of sensory processing than others [25] and has stronger responses to environmental stimuli. Some publications confirm the adaptive function of sensory processing sensitivity (SPS) to environmental stressors [26,27]. Some studies have found positive effects such as better interpersonal skills in highly sensitive individuals [28,29,30]. However, other research attests to the vulnerability induced by high sensitivity, leading to higher levels of anxiety and/or depression (e.g., [31,32,33,34]), perceived stress (e.g., [35,36]), and lower health scores [37].

The literature has confirmed the existence of several components in SPS and their role in the expression of symptoms [30,38,39,40,41,42,43,44]. While the literature often refers to a three-factor model [38], there is some research evidence of four components [39,40]. The first three components include the Ease of Excitation (EOE) and the Low Sensory Threshold (LST), which refer to stimuli that can be experienced as negative. These two components generally tend to increase the level of distress related to general anxiety and depression [30,41]. Their roles may differ with respect to depressive symptomatology and studies reach different conclusions, associating it with either EOE [41] or LST [42]. Aesthetic Sensitivity (AES) refers to one’s deep appreciation of, and connection with, the arts and seems to be a resource allowing, for example, some highly sensitive people to express a greater sense of well-being [43]. It may even play a protective role against depression, to which this component is sometimes negatively correlated [44]. But it may also be a vulnerability factor concerning anxiety in general [30,38,41,42]. Finally, the fourth component, Controlled Harm Avoidance (CHA), is related to the tendency of some highly sensitive individuals to seek to implement certain coping strategies, by being more conscientious, for example [39,40]. However, questions remain about this component, and in particular about its true nature. Described as such, it appears to be a direct consequence of high SPS, rather than one of the contributing factors. In this sense, it could be viewed more as a coping strategy, leading individuals to seek solutions or alternatives to better manage their sensitivity.

In the context of health, studies have found that higher levels of BO and compassion fatigue are correlated to highly sensitive scores for health care workers. Meyerson et al. [20] showed that highly sensitive dentists reported higher levels of BO and lower levels of quality of life than others. Their BO rates were positively associated with EOE and LST and negatively associated with AES. Perez-Chacon et al. [21] pointed out that EOE positively influenced the “emotional exhaustion” component of the BO, while it negatively influenced the “personal fulfilment” component. The latter was positively influenced by the LST. Compassion satisfaction was positively influenced by the EOE and the AES. Specifically, regarding caring and helping professions, compassion satisfaction is described as a protection against BO [14,45] as it reduces its negative effects [21,46]. Furthermore, the literature review [14] suggests that socialisation at work promotes the development of compassionate satisfaction. In this respect, strong social networks would notably reduce the effect of BO [47,48].

While most research on burnout amongst health professions has focused on doctors and nurses [49], there has been little research on other health professions, such as speech and language therapists (SLTs). Brito-Marcelino’s literature review [50] found only six articles on burnout among SLTs, three of which were published before 2000. Yet, like rehabilitation professionals, occupational therapists or physiotherapists, speech-language therapists are also at high risk of BO [51]. Recently, the study by Kasbi et al. [52] showed that 55.5% of Iranian SLTs were reported to have moderate to high BO, mainly due to emotional exhaustion and depersonalisation.

As no study seems to have addressed this topic with the French SLT population, one of the main objectives is to investigate the rate of burnout according to their degree of sensory processing sensitivity (SPS). In line with the literature, we pursue three objectives:To compare the proportion of SLTs expressing high BO with the general population’s rate.To investigate the relation between SLTs’ sociodemographic characteristics and BO scores.To determine the prevalence of sensory processing sensitivity among SLTs and its association with burnout scores.

## 2. Materials and Methods

### 2.1. Sample

Our sample consisted of 602 French-speaking SLTs, all women. Among the respondents, 82.39% (n = 496) are from France and 17.61% (n = 106) from Belgium. Sociodemographic characteristics are provided Table 1. The sample is not representative. In France for example, in 2019, there were 25,607 speech and language therapists: 96.8% were female, 81.12% were self-employed or mixed, 7.33% were hospital-based, and 11.2% were other salaried).

### 2.2. Procedure

The questionnaire was created as a Google Form and distributed via social networks, several SLT Facebook pages, and a SLT welfare page. It was also relayed through private SLT networks and by word of mouth. Participants were invited to respond online and were informed that their responses would be anonymous and confidential. Prior to completing the questionnaire, participants were informed of the objectives of the study and were explicitly asked for their consent to continue the study. To access the questionnaire, participants had to click on “accept and continue” after having read the consent form and given consent to participate. Responses to the Google Form were opened between 20 December 2021 and 28 February 2022. The estimated completion time was approximately 10–15 min.

### 2.3. Measures

The questionnaire consisted of 37 items, excluding the sociodemographic questions.

Burnout (BO) was measured with ProQOL 5th version [53]. In our sample, the internal consistency of burnout (BO) was satisfactory (α = 0.801; 10 items). The response modalities were given from 1 (never) to 5 (very often) on a Likert scale. Items 1, 4, 15, 17 and 29 from the BO subscale were reversed. According to the ProQOL manual, a score of 22 or less corresponds to a low level of BO. A score between 23 and 41 corresponds to a moderate level, and a score of 42 and above corresponds to a high level.

Sensory processing sensitivity was measured using the HSPS-FR [39], which is a self-report questionnaire. The scale consists of 27 items. Internal consistency of the overall scale was excellent (α = 0.918). That of the three components was satisfactory: ease of excitation (EOE) (α = 0.836), low sensory threshold (LST) (α = 0.850), and aesthetic sensitivity (AES) (α = 0.737). That of controlled harm avoidance (CHA) was quite low (α = 0.54), but it does correspond with the literature [39,40]. The response modalities were given from 1 (strongly disagree) to 7 (strongly agree) on a Likert scale. An overall high sensitivity score can be calculated by adding up all of the items. The higher the score, the higher the sensitivity. Within a categorical perspective, Lionetti et al. [23] devised a way of considering three groups according to their degrees of sensitivity: “low sensitivity” (individual’s total scores strictly below 113), “medium sensitivity” (scores between 113 and 137), and “highly sensitive” (scores strictly above 137).

Socio-demographic questions were asked, such as gender (Female; Male; Non-binary; but due to the very low number of responses from male (9) and non-binary (3), we decided to discard these data), age (ages were presented in 10 year increments, starting at 20 years old), country of training (France; Belgium), the number of years of professional practice (strictly less than 5 years; between 5 and 10 years; between 11 and 20 years; strictly more than 20 years) and the practice setting (private practice; medical-social institution; hospital; mixed).

### 2.4. Analyses

Statistical analyses of the questionnaire responses were then carried out using JASP. Firstly, descriptive analyses were performed for BO and HSPS-FR and its subdimensions (EOE, LST, AES, CHA). As the data did not follow a normal distribution, non-parametric tests and Dunn’s post-hoc comparisons were used to measure the differences between the subgroups in our sample, according to socio-demographic characteristics. Then, linear regressions were performed to measure the predictive power and the effects of the variables upon each other.

## 3. Results

### 3.1. Descriptive Results 

According to the low, medium, and high thresholds considered in the literature (Lionetti et al., 2019 [23]), 77.41% of the sample reported moderate or high scores of BO and 50.5% reported high scores of HSPS (Table 2).

Mean scores for BO revealed that all subgroups reported a moderate rate of BO and moderate scores of sensory processing sensitivity (except for the youngest and the ones working in medical-social structures whose scores are higher than 137, revealing a high sensory processing sensitivity) (Table 3).

### 3.2. Study of the Relationship between Socio-Demographic Characteristics, BO Scores, and SPS Scores

Median scores, inter-quartile ranges, Mann–Whitney and Kruskal–Wallis tests, and *p* values are given Table 3.

The results showed that there was a significant difference according to age category (*p* = 0.036) and number of years of practice (*p* = 0.027). Dunn’s post-hoc comparisons were carried out. In the first case, Dunn’s test showed that the difference was between the “20–29 years” group and the “30–39 years” group (*p* = 0.003). In the second case, Dunn’s test showed that the significant difference was between participants with “less than 5 years of practice” and those with “between 11 and 20 years of practice” (*p* = 0.003).

There were also significant differences with these two variables and SPS scores (*p* = 0.003 and *p* = 0.009 respectively). With regard to age, Dunn’s test showed the significant difference was between the oldest group (50 years old and more) and the two youngest groups (20–29 years (*p* < 0.001) and 30–39 years (*p* = 0.005)). These differences are also apparent for the number of years of practice between the ones with the longest service and all the other groups (Table 4). Dunn’s test also showed a difference between respondents who obtained their diploma in 3 years and those who obtained it in 5 years, with a lower BO score and a higher SPS score for those who obtained their diploma in 5 years.

### 3.3. Study of the Relationship between Sensory Processing Sensitivity and BO Scores

The Kruskall–Wallis test showed that SPS was significantly associated with BO scores (*p* < 0.001) (Table 5). Dunn’s post-hoc comparison test revealed that significant differences appeared between the three subgroups (hypo, medium, and highly sensitive) (Table 5).

Linear regression showed that a high sensory processing sensitivity leads to higher BO scores (F(4.597) = 27.363; *p* < 0.001). More specifically, BO was predicted by EOE (t = 5.178) and AES (t = −3.857) (*p* < 0.001) and LST (t = 2.877; *p* = 0.004) (Table 6, r^2^ = 0.155).

## 4. Discussion

This study looked at burnout and sensory processing sensitivity among French speech and language therapists, which is a poorly studied population [50]. We had several objectives: (1) to compare the proportion of SLTs expressing a high BO with the general population’s rate; (2) to investigate the relation between SLTs’ sociodemographic characteristics and BO scores and (3) to determine the prevalence of sensory processing sensitivity among SLTs and the association with burnout scores.

In line with our objectives, our results confirm the higher prevalence of BO among French and Belgian SLTs. Indeed, with over 77% of participants reporting moderate to high burnout, this profession seems to be particularly affected. This rate is even higher than the one (48.6%) reported by the CARPIMKO survey [9]. This could be the consequence of the COVID-19 crisis since the latter is known to have had strong and long-term effects on health and professional quality of life, especially for healthcare workers [54]. But it could equally be due to the burnout scale that we used, which was different from the Maslach Burnout Inventory used in the CARPIMKO survey. This is in line with the sparse results obtained for SLTs in other countries, such as in Italy [51] and in Iran [52] and other studies using MBI in which healthcare professionals obtained above-average scores for all dimensions of burnout (e.g., [55,56]). We thus confirmed the higher prevalence of burnout risk in this population in comparison with the prevalence in the French working population in general. Estimations indicate that around 7% of the 480,000 employees are suffering from work-related psychological distress, i.e., just over 30,000 people [57], and vary from 12% [58] to 34% of the French working population [4]. While these results confirm the vulnerability of speech and language therapists facing burnout, they also prompt questions about the reasons behind such scores. Some answers can already be found in the problematic working conditions of SLTs and healthcare workers in general [10,11,12].

The literature seems to confirm that gender has no influence on BO, e.g., [14,59]. However, differences sometimes appear in the BO forms, where males sometimes report higher depersonalisation [60] or “underchallenged” burnout [56] scores than females. Unfortunately, our study is based on an all-female sample. This is an obvious limitation, directly linked to the type of population studied, to which we must add the tools we used. Indeed, by using the ProQOL, and not the MBI, we were unable to identify the dimensions of the BO for which the women who responded obtained particularly high scores.

We found that the youngest SLTs had the lowest BO scores, although their scores still showed moderate BO. Moreover, the biggest difference related to age appears between respondents aged 20–29 years and those aged 30–39 years, and not with the oldest respondents. This difference is expressed in the sense of a lower BO score for the youngest respondents, which is contradictory to the literature, which states the youngest are more vulnerable to BO [13,14]. However, it should be noted that even though the 30–39-year-olds had higher BO scores, both the 20–29 and 30–39-years-old groups obtained scores revealing a moderate BO; this significant difference reveals a nuance in intensity, but not in categorical nature, of the BO prevalence. This difference between our own results and the literature could be explained by the fact that, since 2013, speech and language therapists have been taking their diploma in 5 years instead of 3. The new diploma incorporates elements relating to quality of working life and analysis of practice, whereas the old diploma was more technical. As a result, the 30–39 age group is mainly made up of speech and language therapists who earned the certificate in 3 years, unlike the youngest group, and may have fewer theoretical and methodological tools to deal with the difficulty of their professional practice. This could explain that they reported higher BO scores than the 20–29-years old group. The oldest SLTs who have also obtained their 3-year diploma could benefit, in line with the literature, from the experience that would enable them to cope better with difficult situations than the younger groups who have also obtained their 3-year diploma.

The proportion of SLTs reporting higher BO scores was also found to be significantly higher among those reporting a higher degree of sensitivity than others and who could be considered as highly sensitive [23]. However, we need to be very cautious about results based on categorical approaches, especially when categories were carried out on populations other than the studied one. Many limits are mentioned towards categorial approach, like the stigmatisation it could induce; but it also has some benefits like favouring “a methodologically differentiated treatment of mental disorders [even if HSPS is not a mental disorder], without prejudging a priori and generically their ontological status” [61]. The use of these categories in this study has only one purpose: to succeed in identifying—whether such a group exists—if the sensory processing sensitivity characteristic leads to more burnout specifically for this group. By confirming the association between SPS and BO scores, this study corroborates recent results obtained with other highly sensitive healthcare professionals [21]. Although the scores obtained by speech therapists mainly indicate a situation of moderate exhaustion rather than high BO, it is clear that the higher their sensitivity, the higher their burnout scores.

Looking specifically at the role and relationship between sensory processing sensitivity components and the different scores of the ProQOL scale, several elements must be underlined. The EOE and AES components played opposing roles in BO. In other words, the more an SLT feels overwhelmed by internal and external stimuli, and the greater awareness they have of the details of their environment, the greater the risk of developing a form of burnout. More specifically, the EOE component generates a higher level of emotional exhaustion (a component of burnout) [21]. Thus, sensory stimuli experienced more intensely (EOE) generate emotional exhaustion that could in turn lead to the expression of a form of BO. The effects of EOE have been widely documented, and they are a strong argument for considering this component as one of the vulnerability factors of high sensory processing sensitivity. Indeed, this component generally tends to increase the level of distress linked to general anxiety and depression [30,41]. Similarly, an increased awareness of details relating to the LST (low sensory threshold) component leads to greater fatigue in the most sensitive individuals, as their attention is more often on high alert. This component appears to play a vulnerabilising role, as does EOE.

## 5. Conclusions and Limitations

While high sensitivity due to the predominance of one or both of these components seems to have rather negative consequences for highly sensitive individuals, there is less consensus in the literature about the effects of the aesthetic component. It is sometimes presented as a protective factor [20], and sometimes as a risk factor [21]. Our results seem to be consistent with those of Meyerson and colleagues, who emphasised its protective role against BO. It could even play a protective role against depression, to which this component is sometimes negatively correlated [41].

From a perspective applied to the quality of professional life of speech language therapists, possible courses of action could be envisaged to reduce burnout among these heath workers, particularly those with a high level of sensory processing sensitivity. We propose three avenues for consideration in relation to the training courses that could be offered to them. The first relates to coping strategies likely to provide better protection against the effects of burnout and/or over-sensitivity to stimuli likely to be experienced negatively. The second is to familiarise them with the tools and methods associated with the practice of mindfulness, so that they are better able to stay in the present moment and become aware of the stimuli they are experiencing. Finally, the third relates to the development of aesthetic sensitivity, which appears to protect against burnout and depression, and could help to increase the feeling of well-being.

The study has a number of limitations, some of which have already been mentioned, such as the fact that the sample was made up entirely of women. There may also have been some selection bias. We collected our data partly via social networks, and in particular on a Facebook page dedicated to the quality of life of speech therapists and the prevention of burnout. In making this selection, we took the risk of recruiting speech therapists who felt particularly affected by this syndrome. This recruitment may also have been biased by the presentation of the questionnaire, indicating that it concerned hypersensitivity. Thus, only speech and language therapists with an interest in the subject may have been more likely to have responded. This could also explain the high scores obtained on the hypersensitivity scale. Speech and language therapists who responded to our questionnaire may also have known about hypersensitivity, or even already considered themselves to be hypersensitive. It is therefore possible that they filled in the scales in this way. In addition, media exposure of the concept has led to a certain valorisation of hypersensitivity. For some people, being hypersensitive now has a positive connotation, especially as the concept is regularly discussed in the media.

## Figures and Tables

**Table 1 ijerph-20-07132-t001:** Sociodemographic characteristics of the sample according to respondents’ country of residence (%).

	*France*	*Belgium*
*Age*		
20–29	19.96	35.85
30–39	32.46	41.51
40–49	27.62	16.98
50 and +	19.96	5.66
Total	100.00	100.00
*Duration of the formation before certification*		
3 years	8.87	75.47
4 years	72.78	10.38
5 years	18.35	14.15
Total	100.00	100.00
*Number of years of professional practice*		
Less than 5 years	18.55	29.24
5 to 10 years	21.57	26.42
11 to 20 years	30.24	32.08
More than 20 years	29.64	12.26
Total	100.00	100.00
*Professional practice*		
Private practice	5.04	7.55
Medical-social structure	81.65	83.96
Hospital	3.23	1.89
Mixed	10.08	6.60
Total	100.00	100.00

**Table 2 ijerph-20-07132-t002:** Descriptive results according to thresholds (low, medium, high) fixed for the scales.

		Low Scores	Medium Scores	High Scores
BO	N	136.00	464.00	2.00
	%	22.59	77.08	0.33
	Mean	19.01	30.27	44.00
	SD	2.70	4.63	1.41
	Median	20.00	30.00	44.00
	IQR	4.00	7.00	1.00
SPS	N	138.00	160.00	304.00
	%	22.92	26.58	50.50
	Mean	96.03	125.97	156.12
	SD	12.40	7.41	12.71
	Median	98.00	126.00	155.00
	IQR	17.75	13.00	19.00

**Table 3 ijerph-20-07132-t003:** Characteristics of the sample, descriptive data, with means, SD, median and IQR, Kruskall–Wall and Mann–Whitney tests for BO scores and SPS scores.

Variables			BO						SPS					
	** *N* **	**%**	** *Mean* **	** *SD* **	** *Median* **	** *IQR* **			** *Mean* **	** *SD* **	** *Median* **	** *IQR* **		
*Gender*														
Female	602	100.00	27.77	6.42	28.00	9.00			134.33	26.95	138.00	40.75		
	** *N* **	**%**	** *mean* **	** *SD* **	** *median* **	** *IQR* **	** *W* **	** *p* **	** *mean* **	** *SD* **	** *median* **	** *IQR* **	** *W* **	** *p* **
*Country of training*														
Belgium	106	17.61	28.91	6.31	29.00	9.00	3.002	0.083	135.81	28.11	138.00	43.50	27.259	0.550
France	496	82.39	27.53	6.42	28.00	9.00			134.01	26.72	138.00	39.00		
*Rank-Biserial Correlation* = 0.107								*Rank-Biserial Correlation* = 0.037						
*SE Rank-Biserial Correlation* = 0.062								*SE Rank-Biserial Correlation* = 0.062						
	** *N* **	**%**	** *mean* **	** *SD* **	** *median* **	** *IQR* **	***χ*2** ***stat***	** *p* **	** *mean* **	** *SD* **	** *median* **	** *IQR* **	***χ*2** ** *stat* **	** *p* **
*Age*														
20–29	137	22.76	26.57	6.27	26.00	9.00	8.570	0.036	138.62	27.45	140.00	37.00	12.917	0.005
30–39	205	34.05	28.65	6.38	29.00	9.00			136.19	26.28	139.00	38.00		
40–49	155	25.75	27.64	6.47	29.00	9.00			133.12	27.04	138.00	42.00		
50 and +	105	17.44	27.84	6.45	27.00	8.00			126.88	26.20	129.00	41.00		
*Duration of the formation before certification*														
3 years	124	20.60	28.77	6.35	28.50	9.00	5.870	0.053	129.35	28.96	131.00	45.00	5.315	0.070
4 years	372	61.80	27.74	6.43	28.00	9.00			135.12	26.56	139.00	39.00		
5 years	106	17.60	26.73	6.34	27.00	8.00			137.37	25.35	138.00	36.75		
*Number of years of professional practice*														
Less than 5	123	20.43	26.59	6.31	26.00	9.00	9.161	0.027	136.89	27.36	138.00	39.00	11.642	0.009
5 to 10	135	22.43	27.79	6.45	28.00	9.00			136.80	25.88	139.00	35.50		
11 to 20	184	30.56	28.67	6.38	29.00	9.00			136.28	26.90	139.00	38.50		
More than 20	160	26.58	27.64	6.43	28.00	9.00			128.04	26.84	130.50	44.00		
*Professional practice*														
Private practice	494	82.06	27.83	6.39	26.00	8.00	1.930	0.587	134.17	26.44	138.00	34.00	0.409	0.938
Medical-social	33	5.48	26.70	5.92	28.00	9.00			137.58	23.87	138.00	38.00		
Hospital	18	2.99	26.56	7.10	26.50	9.00			134.33	29.40	138.50	45.75		
Mixed	57	9.47	28.28	6.79	29.00	12.00			133.81	32.41	137.00	52.00		

**Table 4 ijerph-20-07132-t004:** Dunn’s post-hoc comparison—age categories and number of years of professional practice.

BO Scores					
**Dunn’s Test for Age Categories**		** *z* **	** *Wi* **	** *Wj* **	** *p* **
20–29	30–39	−2.927	267.766	323.873	0.003
20–29	40–49	−1.692	267.766	302.245	0.091
20–29	50 and +	−1.463	267.766	300.733	0.143
30–39	40–49	1.170	323.873	302.245	0.242
30–39	50 and +	1.110	323.873	300.733	0.267
40–49	50 and +	0.069	302.245	300.733	0.945
**Dunn’s test for duration before certification**		** *z* **	** *Wi* **	** *Wj* **	** *p* **
3 years	4 years	1.119	323.649	303.497	0.263
3 years	5 years	2.396	323.649	268.580	0.017
4 years	5 years	1.825	303.497	268.580	0.068
**Dunn’s test for number of years of professional practice**		** *z* **	** *Wi* **	** *Wj* **	** *p* **
Less than 5	5 to 10	−1.719	265.902	303.137	0.086
Less than 5	11 to 20	−3.014	265.902	326.891	0.003
Less than 5	More than 20	−1.554	265.902	298.284	0.120
5 to 10	11 to 20	−1.207	303.137	326.891	0.228
5 to 10	More than 20	0.239	303.137	298.284	0.811
11 to 20	More than 20	1.523	326.891	298.284	0.128
**SPS scores**					
**Dunn’s test for age categories**		** *z* **	** *Wi* **	** *Wj* **	** *p* **
20–29	30–39	1.034	331.062	311.217	0.301
20–29	40–49	1.755	331.062	295.281	0.079
20–29	50 and +	3.455	331.062	253.138	<0.001
30–39	40–49	0.861	311.217	295.281	0.389
30–39	50 and +	2.783	311.217	253.138	0.005
40–49	50 and +	1.917	295.281	253.138	0.055
**Dunn’s test for duration before certification**		** *z* **	** *Wi* **	** *Wj* **	** *p* **
3 years	4 years	−1.947	271.121	306.233	0.052
3 years	5 years	−2.143	271.121	320.429	0.032
4 years	5 years	−0.741	306.233	320.429	0.458
**Dunn’s test for number of years of professional practice**		** *z* **	** *Wi* **	** *Wj* **	** *p* **
Less than 5	5 to 10	0.251	320.630	315.189	0.802
Less than 5	11 to 20	0.356	320.630	313.427	0.722
Less than 5	More than 20	2.834	320.630	261.528	0.005
5 to 10	11 to 20	0.089	315.189	313.427	0.929
5 to 10	More than 20	2.640	315.189	261.528	0.008
11 to 20	More than 20	2.761	313.427	261.528	0.006

**Table 5 ijerph-20-07132-t005:** Dunn’s post-hoc comparison for the three groups on burnout scores according to sensory processing sensitivity degree (hypo, medium, and highly sensitive).

Factor	*χ2 Stat*	*p*			
SPS-categories	68.859	<0.001			
		** *z* **	** *Wi* **	** *Wj* **	** *p* **
Highly sensitive	Hypo sensitive	8.287	352.275	204.493	<0.001
Highly sensitive	Medium sensitive	3.747	352.275	288.697	<0.001
Hypo sensitive	Medium sensitive	−4.172	204.493	288.697	<0.001

**Table 6 ijerph-20-07132-t006:** Coefficients of linear regression for BO with SPS components.

Model		Unstandardised	SE	Standardised	t	*p*
H₀	(Intercept)	27.774	0.262		106.150	<0.001
H₁	(Intercept)	21.135	1.660		12.728	<0.001
	EOE	0.168	0.033	0.321	5.178	<0.001
	LST	0.101	0.035	0.175	2.877	0.004
	AES	−0.214	0.056	−0.176	−3.857	<0.001
	CHA	−0.032	0.101	−0.015	−0.320	0.749

## Data Availability

All the data used for this research are available upon request to the corresponding author.

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
