# Peer review of "Study of the Relationship between Sensory Processing Sensitivity and Burnout Syndrome among Speech and Language Therapists"

_ijerph, 2023, doi:10.3390/ijerph20237132_

Round 1

Reviewer 1 Report

Comments and Suggestions for Authors

Dear Authors,

I read your work entitled “Influence of sensory processing sensitivity on burnout syndrome among French speech and language therapists” and here I enclose my recommendations to you:

1.     This work needs some editing for its English. The syntax of the sentences in some case does not leads to any kind of meaning.

2.     The “Results” it is not readers friendly and some analysis issues are raised. Specifically, the Authors state in their “Methods section” that “As the data did not follow a normal distribution non-parametric tests were used …” from this it is expected in the “Results section” to see Medians and IQR instead of Means and SDs so all Table must be changed accordingly. The Authors report their data in a manner that does not help the reader, for example in the “Influence of sociodemographic characteristics” subchapter they stated that they have used a Kruskal-Wallis test and they have reported t values. We also do not see Mann-Whitney test between pairs. I strongly suggest the Authors to address such issues.

3.     The Introduction, Methods and Discussion section is very good, and I congratulate the Authors for that.

4.     Please add limitations to this study.

Thank you.

Comments on the Quality of English Language

 Minor editing of English language required

Author Response

Thank you for your reviewing and very relevant comments. We propose here an answer to each one (in bold).

I read your work entitled “Influence of sensory processing sensitivity on burnout syndrome among French speech and language therapists” and here I enclose my recommendations to you:

  1. This work needs some editing for its English. The syntax of the sentences in some case does not leads to any kind of meaning.
  • The article was proofread by a native English speaker. So, we are not sure we understand the comment. Perhaps you could tell us exactly which sentences are causing concern so that we can reword them if necessary.
  1. The “Results” it is not readers friendly and some analysis issues are raised. Specifically, the Authors state in their “Methods section” that “As the data did not follow a normal distribution non-parametric tests were used …” from this it is expected in the “Results section” to see Medians and IQR instead of Means and SDs so all Table must be changed accordingly.
  • Thanks for this very important comment. That was a mistake and something we forgot to provide. We added the information related to medians and IQR with statistics and p value (Table 2).
  1. The Authors report their data in a manner that does not help the reader, for example in the “Influence of sociodemographic characteristics” subchapter they stated that they have used a Kruskal-Wallis test and they have reported t values.
  • The values have been updated (Table 3)
  1. We also do not see Mann-Whitney test between pairs. I strongly suggest the Authors to address such issues.
  • Dunn’s post hoc comparisons have been carried out (Table 4 et 5)
  1. The Introduction, Methods and Discussion section is very good, and I congratulate the Authors for that.
  • Thanks for this very positive and gratifying feedback.
  1. Please add limitations to this study.
  • We added the following paragraph in the section “conclusions and limitations” :

“The study has a number of limitations, some of which have already been mentioned, such as the fact that the sample was made up entirely of women. There may also have been some selection bias. In addition, we collected our data partly via social networks, and in particular on a network page dedicated to the quality of life of speech therapists and the prevention of burnout. In making this selection, we took the risk of recruiting speech therapists who felt particularly affected by this syndrome. This recruitment may also have been biased by the presentation of the questionnaire, indicating that it concerned hypersensitivity. Thus, only speech and language therapists with an interest in the subject may have been more likely to have responded. This could also explain the high scores obtained on the hypersensitivity scale. Speech and language therapists who responded to our questionnaire may also have known about hypersensitivity, or even already considered themselves to be hypersensitive. It is therefore possible that they filled in the scales in this way. In addition, media exposure of the concept has led to a certain valorisation of hypersensitivity. For some people, being hypersensitive now has a positive connotation, especially as the concept is regularly discussed in the media.”

Reviewer 2 Report

Comments and Suggestions for Authors

Dear Author,

First I would like to congratulate you on the choice of the topic. The study on the whole makes a good impression and may have scientific significance for future studies. I have a few suggestions which could improve your study.

You used the word "influence" in the Title and the whole text. I think this type of study and the statistical analyses didn't measure the influence, maybe it is better to use other words like "association, associated or relationship...".

Also, the Title is "Influence on sensory processing sensitivity on burnout syndrome among French speech and language therapist", but there are three objectives of this study. I think you could arrange the title so it could better fit the objectives and type of the study.

In the Introduction (paragraph 4, line 62) you wrote "many studies...", and after this sentence, there are no citations. Please add it.

I think that paragraph Procedure should be before the Measures.  Please describe the sample better. You could state how many SLTs exist in France and their demographic characteristics if you have them. Is the sample representative or not?

In addition, you wrote, "Our results confirmed the significant risk of BO among Fench SLT". As I earlier wrote, this type of study and analysis didn't investigate the risk factor or risk. Maybe it is better to write, "Our results confirm the higher prevalence of BO ..."

In the Discussion, you explain a possible reason for this higher prevalence. But having in mind that literature highlighted that women are more prone to BO and that your sample is all women, this could also be one of the reasons. Please add it and discuss it. You need to discuss about age difference in BO and add possible reasons for these results and differences.

Limitations and advantages of the study are missing. Please add it.
